# STRATEGY-CENTRIC SYNTHESIS: CONNECTING BILLIONS OF IMAGE-TEXT PAIRS TO HIGH-QUALITY VISUAL INSTRUCTION DATA.

## ABSTRACT

Vision-Language Models (VLMs) have demonstrated remarkable generalization across tasks by aligning visual and linguistic representations. High-quality visual instruction data is critical for enhancing the performance of Vision-Language Models. However, current visual instruction tuning datasets, which are primarily derived from past visual tasks, have several limitations. For instance, the range of question types is often restricted and closely tied to the original visual tasks. Furthermore, image diversity is limited, as images collected for various specialized vision tasks clearly fail to adequately represent real-world user queries. Additionally, previous instruction datasets tend to lack complexity, focusing on single tasks like captioning or OCR, which makes it challenging to train models for more complex, multi-skill scenarios. To address these limitations, we propose a novel paradigms called strategy-centric synthesis: automatically synthesizing high-quality instruction data from large-scale image-text pairs. First, we employ an efficient heuristic method to select high-quality, complex images from DataComp-1B image-text pairs. Carefully crafted prompts and these images are fed to VLMs to extract high-quality query strategies and generate corresponding image descriptions. These descriptions are subsequently used to retrieve images aligned with specific questioning strategies. Finally, the retrieved images and their matching strategies are used to synthesize high-quality instructional data. Our experiments indicate that with continued instruction fine-tuning via LoRA on only 3,000 newly synthesized data samples, 0.45% of the LLAVA-1.5 instruction tuning dataset, the model significantly outperforms the original LLAVA-1.5-7B across multiple benchmarks, thereby demonstrating the effectiveness of our approach.

## 1 INTRODUCTION

Multimodal Large Language Models (MLLMs)(Liu et al., 2024c; Zhu et al., 2023; Li et al., 2023a; Dai et al., 2023; Tong et al., 2024; Wang et al., 2024) have demonstrated strong cross-task generalization in recent years. Typical architectures consist of a pre-trained visual backbone (Radford et al., 2021; Sun et al., 2023) for encoding visual features, a pre-trained LLM (Touvron et al., 2023; Chiang et al., 2023) to interpret user instructions and generate responses, and a vision-language cross-modal connector to align visual encoder outputs with the language model. Training an instruction-following LMM typically follows a two-stage protocol. First, the pretraining stage leverages image-text pairs to align visual features with the language model's word embedding space. Second, the visual instruction tuning stage fine-tunes the model on visual instructions, enabling it to handle diverse user requests that involve visual content. For the pretraining stage, the abundance of image-text pairs accumulated from prior research means that data is not a significant bottleneck. However, in the visual instruction tuning stage, there is a clear lack of sufficient high-quality instruction data. Previous approaches have transformed data from previous visual task datasets using templates (Xu et al., 2022), manual annotations (Xu et al., 2023), language models (Liu et al., 2024c; Tong et al., 2024), or vision-language models (Zhao et al., 2023; Wang et al., 2023; Chen et al., 2023b) to generate instruction data.

However, these datasets exhibit several clear limitations:

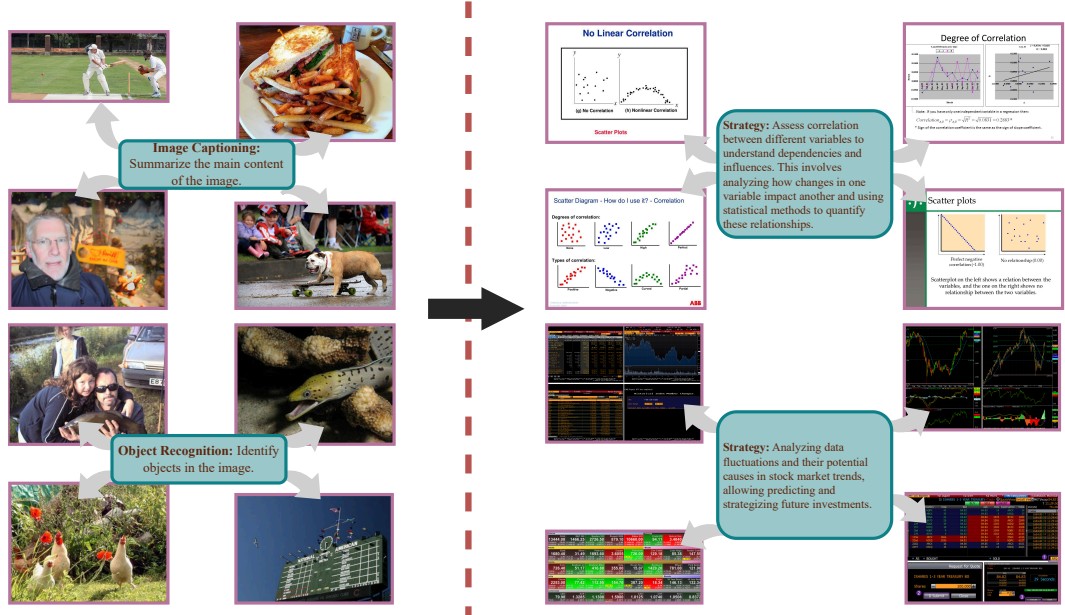

Figure 1: A comparison of two different paradigms for constructing instruction data. The left side illustrates the previous instruction data centered on vision-based tasks, in which instruction data is generated through templates or rewriting. In contrast, the right side is centered on strategies that guide the VLM model in synthesizing high-quality queries for a category of images with shared features.

1. **Lack of Diversity**: Since most instruction datasets are derived from previous vision tasks, these datasets face limitations in diversity, specifically in the following aspects: (1) Limited Question Types: The types of questions are highly correlated with the original tasks, and the categories of tasks themselves are limited. For instance, Xu et al. (2023) compiled nearly all past vision tasks but only managed to obtain 200+ diverse vision-language tasks, resulting in a limited variety of question types. (2) Limited Image Distribution: The images collected for specialized vision tasks are clearly insufficient to cover the distribution of real-world user queries. (2) Limited Variety of Prompt Templates Used for Synthesis: Past work has often used a static template to prompt VLMs (vision-language models) to synthesize instruction data for different images, such as detailed captions (Chen et al., 2023b) or complex reasoning tasks (Liu et al., 2024c; Chen et al., 2024), which restricts the full potential of the VLMs.

2. **Lack of Complexity**: In most previous queries, only one basic visual ability, such as captioning or OCR, is typically involved. However, real-world scenarios often require a combination of multiple abilities to resolve queries. Fine-tuning on previous instruction data does not adequately teach VLMs to master a combination of multiple capabilities.

3. **Mismatch Between Images and Prompt Templates**: Each image has an optimal questioning approach, but previous synthesis methods have not sufficiently considered this. For instance, simple images selected from image caption datasets are sometimes forced into generating complex reasoning instruction data (Liu et al., 2024c; Chen et al., 2024), which can lead the model to produce divergent questions rather than high-quality reasoning problems.

To address the aforementioned issues, we propose a novel paradigms called strategy-centric synthesis. Before presenting our method, we introduce the query strategy for visual instructions, which refers to a general questioning perspective applicable to images with some shared characteristics. As illustrated in Figure 1, the strategy can be used to guide the question synthesis for a category of images. It is evident that query strategies are more fine-grained than foundational visual task descriptions, making them more suitable for handling complex scenarios. In fact, various basic visual

tasks can be viewed as specific query strategies within our method. For instance, an OCR-related task can correspond to the strategy: designing questions about text recognition in the image.

Our method centers around strategies and consists of two primary components: (1) automated strategy mining from seed images using visual language models, and (2) strategy-guided multimodal instruction synthesis. First, we introduce a heuristic and efficient approach for selecting high-quality, complex images from large-scale image-text pairs by leveraging domain-specific visual keywords to filter image captions and identify relevant images. Using this approach, we construct a diverse and complex seed image library from the recaptioned DataComp-1B dataset (Li et al., 2024). Carefully designed prompts are then applied to mine high-quality query strategies and generate corresponding image type descriptions. After eliminating redundant strategies, we use the image type descriptions to retrieve matching images, guide question generation based on the associated strategies, and produce detailed, step-by-step answers. Finally, a self-reflection step is implemented to evaluate the quality of the synthesized instructions.

Our core contributions are as follows:

1. The proposed strategy-centric data synthesis approach effectively addresses several clear limitations observed in existing visual instruction datasets. By integrating query strategies into the synthesis process, we enhance the **diversity** of both prompt templates and question types. Moreover, these strategies guide visual language models to generate higher-quality, **complex** instruction queries at a finer granularity. During synthesis, retrieving matching images based on the strategy's corresponding image type descriptions also significantly mitigates the **mismatch** between images and prompt templates.

2. We introduce an automated strategy mining approach, starting with a heuristic retrieval method to efficiently collect images suitable for generating complex queries. Using these seed images, we prompt the visual language model to generate query strategies. Our methods connects billions of image-text pairs to high-quality visual instruction data, providing potential scalability for high-quality data synthesis.

3. After continued LoRA instruction tuning using only 3k synthesized data samples, 0.45% of the LLAVA-1.5 instruction tuning dataset, the model significantly outperforms the original LLAVA-1.5-7B across multiple benchmarks.

## 2 RELATED WORK

Multimodal Large Language Models (MLLMs) have made significant strides in recent years, driven by the success of Large Language Models (LLMs). Typical architectures consist of a pre-trained visual backbone for encoding visual features, a pre-trained LLM to interpret user instructions and generate responses, and a vision-language cross-modal connector to align visual encoder outputs with the language model. Models such as LLaVA (Liu et al., 2024c;a) and MiniGPT-4 (Zhu et al., 2023) have demonstrated strong cross-task generalization. mPLUG-Owl (Ye et al., 2023; 2024), Shikra(Chen et al., 2023a), and KOSMOS-2 (Peng et al., 2023) have introduced novel data types and training methods, such as grounding data, aimed at reducing hallucinations and improving the grounding capabilities of LLMs. LLaVA-NeXT (Liu et al., 2024b) has significantly enhanced visual perception by utilizing dynamic resolution techniques, while Cambrain1 (Tong et al., 2024) has improved model robustness through visual encoder routing. Recently, Luo et al. (2024); Xie et al. (2024); Zhou et al. (2024) have combined diffusion models with LLMs to enhance both the generative and understanding capabilities of MLLMs.

Training an instruction-following LMM typically follows a two-stage protocol. First, the vision-language alignment pretraining stage leverages image-text pairs to align visual features with the language model's word embedding space. Second, the visual instruction tuning stage fine-tunes the model on visual instructions, enabling it to handle diverse user requests that involve visual content. For the pretraining stage, the abundance of image-text pairs accumulated from prior research means that data is not a significant bottleneck. However, in the visual instruction tuning stage, there is a clear lack of sufficient high-quality instruction data. Previous approaches have transformed data from single-task visual datasets using templates (Xu et al., 2022), manual annotations (Xu et al., 2023), language models (Liu et al., 2024c; Tong et al., 2024), or vision-language models (Zhao et al., 2023; Wang et al., 2023; Chen et al., 2023b) to generate instruction data. Unlike these datasets,

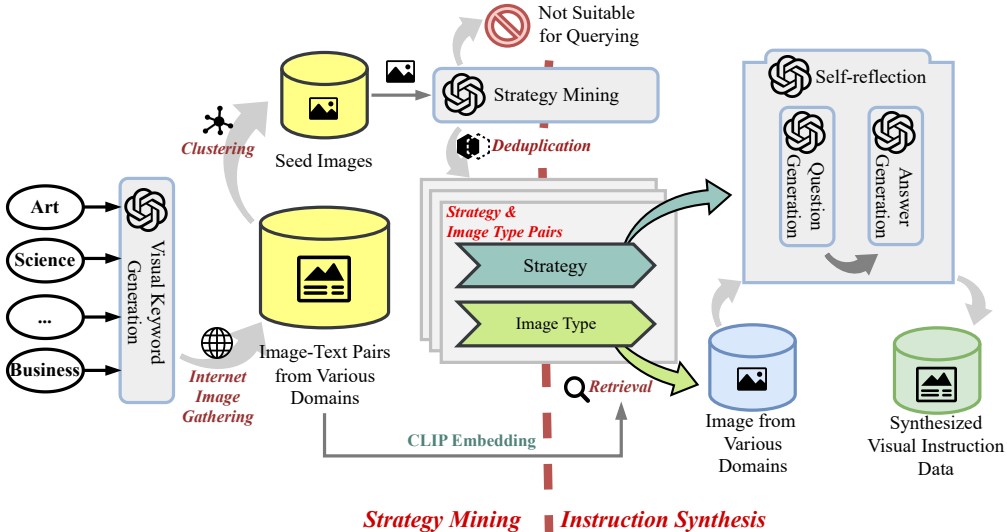

Figure 2: Our method centers around strategies and involves two main components: (1) automated strategy mining from seed images using visual language models, and (2) strategy-guided multimodal instruction synthesis. First, we build a diverse and complex seed image library using recaptioned DataComp-1B pairs. We then apply carefully designed prompts to mine high-quality query strategies and corresponding image type descriptions from these images. After deduplicating redundant strategies, we use the image type descriptions to retrieve matching images and guide question generation with the corresponding strategies, producing step-by-step answers. Finally, a self-reflection step evaluates the quality of the synthesized instructions.

where images are primarily collected for specific visual tasks and thus have biased distributions, our approach is based on billions of image-text pairs, allowing us to construct more diverse and realistic instruction data. Moreover, we believe that different types of images have their own optimal questioning strategies. In contrast to methods like ShareGPT4V (Chen et al., 2023b) and ALLaVA (Chen et al., 2024), which often apply fixed prompts to all images, we mine large-scale questioning strategies and dynamically adapt them to suit each image type, thereby significantly improving the quality of instruction data at the case level.

## 3 METHOD

As shown in Figure 2, our approach consists of two key components: (1) the automated mining of high-quality strategies from seed images using visual language models, and (2) strategy-guided multimodal instruction data synthesis.

### 3.1 AUTOMATED STRATEGY MINING

Manually annotating query strategies is prohibitively costly and time-consuming. In practice, we have found that with carefully designed prompts, powerful visual language models can mine high-quality potential query strategies directly from representative images. This insight led us to develop the automated mining method.

**High-Quality Seed Image Library Construction** The construction of a high-quality seed image library focuses on two main criteria: (1) Diversity: Traditional instruction datasets built from individual academic tasks often contain many homogeneous images. To cover a broader range of image types, we leverage recaptioned DataComp-1B image-text pairs (Li et al., 2024). We start by identifying common real-world domains, such as science, medicine, and business. For each domain, we generate a series of visual keywords using large language models (LLMs) through prompts, then use these keywords to filter captions from the large-scale image-text pairs, allowing us to acquire matching images. (2) Complexity: We assume that the more keywords in the caption of the image,

*Given the current image, analyze the specific features and elements, and identify potential questioning angles in real-life scenarios. Then, devise detailed and general strategies for generating questions. Finally, provide the general types of images to which these strategies are applicable.*

*Please respond in the following format:*

*Analysis:[ Analyze the specific features and elements of the image and identify potential questioning angles in real-life scenarios.]*

*Strategies:[ List detailed and general questioning strategies. Don't need the example question.]*

*Images:[Describe types of images where these strategies would be applicable.]*

Figure 3: Prompt used for strategy mining.

the higher the upper bound of complexity for formulating questions. Thus, we select images that better reflect the intricacies of real-world scenarios by the number of visual keywords present in the captions. According to the above principles and methods, we collect images from different domains, cluster them using embedding representations and select representative images as seed images.

**Strategy Mining** Using these high-quality seed images, we employ carefully designed prompts to guide the visual language model in strategy mining. Specifically, we first prompt the model to assess whether the image is suitable for querying, then extract general and detailed query strategies from multiple perspectives based on the image. These strategies provide questioning perspectives for images with certain similar features. Finally, the model generate the types of images to which these strategies could be applied. The detailed prompt can be found in Figure 3.

**Deduplication of Mined Strategies** Despite the use of representative images from various domains, the mined strategies may still exhibit redundancy. To address this, we perform deduplication. We compute embeddings for the strategies, calculate pairwise cosine similarities, and filter them based on a threshold determined through empirical testing.

## 3.2 Strategy-Guided Multimodal Instruction Synthesis

Building on the strategies and their corresponding image types, we introduce a comprehensive dynamic strategy-driven approach for synthesizing multimodal instructions.

**Candidate Image Retrieval** Given a strategy and the corresponding image types, the process begins with the retrieval of images that match the image types from our previously constructed multi-domain image library. Specifically, we leverage a CLIP model to transform the image type descriptions and the multi-domain image library into the embedding space for similarity matching.

**Question Generation** Once the matched images have been retrieved, the corresponding strategy will guide the visual language model to generate high-quality questions. The detailed prompt can be found in Figure 5. These questions essentially represent the concretization of the strategy given the current image. By increasing the number of images retrieved for each strategy, we can easily scale the dataset to a larger size. Therefore, our approach has significant scalability advantages.

**Step-by-Step Answer Generation** Following the question generation, the visual language model will be prompted to generate detailed, step-by-step answers. These answers are crafted to not only address the query but also to provide comprehensive explanations and the reasoning process needed

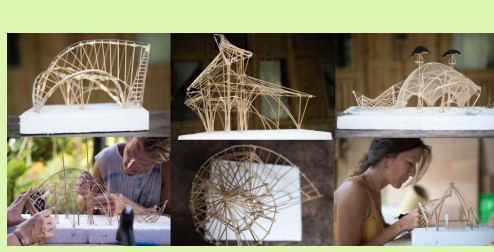

**Category:** Technology
**Question:** What materials are used in creating the objects shown in the image, and what does their craftsmanship suggest about the technological advancements and cultural influences in their production?
**Answer:** The objects in the image are created using thin strips of bamboo or similar lightweight wood, which are carefully bent and joined to form intricate structural models. The use of bamboo, a sustainable material, suggests an emphasis on eco-friendly construction practices. The craftsmanship involved in bending and joining the bamboo strips points to a high level of skill and precision, …

**Strategy:** Investigate the materials used and the craftsmanship involved in creating the object and its implications on trade, technology, and cultural exchange.

Figure 4: A randomly sampled case. More cases can be found in the appendix.

*Formulate a high−quality question based on the image, and refer to the following question strategies : {}. Then generate a detailed, step−by−step solution, and reflect on whether the answer is correct. Ensure your response is valid JSON. Do not include any extra text, explanations, or formatting. Output only the JSON object, in the exact format below, so it can be directly loaded using 'json.load':*

```
{{
    "question": "Formulate a high−quality question based on the
    image, referencing the question strategies mentioned above.",
    "answer": "Provide a detailed, step−by−step solution to the
    question proposed above.",
    "evaluation_answer": "Evaluate the correctness of the answer,
    only output 'yes' or 'no'."
}}
```

Figure 5: Prompt used for instruction synthesis using a given strategy. {} serves as a placeholder for the strategy.

to solve the problem. The step-by-step nature of these answers is critical in ensuring clarity, particularly for complex visual tasks that require multi-step reasoning or the integration of multiple fundamental visual tasks.

**Self-Reflection for Quality Evaluation** To ensure the quality and accuracy of the synthesized instructions, we implement an additional step where the model reflects on its own outputs. This self-reflection phase encourages the model to assess the correctness and completeness of its answers, identifying any potential errors in reasoning. We directly filter out examples that the model identifies as incorrect.

# 4 EXPERIMENT

## 4.1 IMPLEMENTATION DETAILS

| Domain | Tech | Art | Business | Medicine | Science | Sociology |
|---|---|---|---|---|---|---|
| Number | 66,388 | 86,585 | 48,735 | 48,503 | 163,965 | 82,277 |

Table 1: Number of images downloaded across different domains.

Our dataset construction primarily builds on the recaptioned dataset (Li et al., 2024), which employs a vision-language model (VLM) to recaption billions of text-image pairs. We began by identifying several distinct domains: Tech, Art, Business, Medicine, Science, and Sociology. With the help of GPT-4 and manual verification, we curated a set of visualization-related keywords for each domain, which can be found in the Appendix. We hypothesize that the more keywords in the caption of the image, the greater its potential for generating complex questions. Therefore, if a caption contains more than four visualizable keywords and the image resolution exceeds (336, 336), we download and save the image. By traversing billions of text-image pairs, we successfully downloaded a diverse set of images across these domains, as detailed in Table 1.

Next, we applied k-means clustering techniques to these images, grouping them into 1,00 clusters for each domain. It is important to emphasize that more clusters can be set here; however, due to our limited budget, we only need a small number of images, so we choose to use a relatively small clustering cluster. For each cluster, we selected the image closest to the centroid as the representative image, while the remaining images were reserved for future retrieval. Using the GPT-4o visual language model, we generated general and detailed questioning strategies from multiple perspectives for each image. Using regular expressions, we extracted approximately 2,000 strategies and their corresponding image type descriptions from the response. We then applied semantic deduplication to these strategies, utilizing OpenAI's "text-embedding-3-small" model to obtain their embedded representations. By setting a cosine similarity threshold of 0.65, we reduced the set to about 1000 unique questioning strategies.

When retrieving images using image type descriptions, we employed the CLIP model (Li et al., 2024), which was trained on a large corpus of internet images. Given that different strategies might correspond to similar image types, some overlap in the retrieved images was anticipated. To address this, we randomly selected one image from the top k (k=5) retrieval results to increase diversity. Finally, considering budgetary constraints, we synthesized three data cases per strategy using the GPT-4o model. Therefore, the final synthesized dataset consists of 3k instances, which is approximately only 0.45% of the original 665k instruction fine-tuning data. The code and data will be open-sourced.

## 4.2 EXPERIMENTAL SETUP

We selected the popular instruction-tuned LLaVA-v1.5-7B (Liu et al., 2024a) model as our baseline and adopted the LoRA technique for further instruction tuning. There are two reasons for this choice: (1) Due to limited budget and computational resources, the scale of our synthetic dataset is relatively small, making it challenging to perform full-scale instruction tuning from scratch. (2) The newly constructed instruction data is often more complex, involving combinations of multiple sub-tasks, which is more suitable for continued learning in a model that already possesses foundational capabilities. The format of the synthetic data is parsed to be consistent with the original instruction dataset of LLaVA-1.5 (Liu et al., 2024a). Additionally, to ensure reproducibility, all hyperparameters used during the LoRA fine-tuning process are kept identical to the original script (Liu et al., 2024a), as detailed in the appendix.

We evaluated the model on multiple mainstream benchmarks: ChartQA (Masry et al., 2022), MME(Fu et al., 2023) MMBench (Liu et al., 2023), MMMU(Yue et al., 2024), POPE (Li et al., 2023b) , ScienceQA IMG (Lu et al., 2022) , TextVQA (val)(Singh et al., 2019) ,VizWiz (Gurari et al., 2018), DocVQA (Mathew et al., 2020). We utilized LLMs-Eval (Bo Li* & Liu, 2024) as the evaluation tool and, to ensure reproducibility, maintained all evaluation parameters at their default settings without any modifications.

| Method | Language Model | ChartQA | MME$^P$/MME$^C$ | MMB | MMMU | POPE | SQA(img) | TextVQA (val) | VizWiz (val) | DocVQA |
|---|---|---|---|---|---|---|---|---|---|---|
| BLIP-2 | FLAN-T5 | - | 1293.8 / 290.0 | - | - | - | 61.0 | - | 19.6 | - |
| InstructBLIP | Vicuna-7B | - | - / - | 36.0 | - | - | 60.5 | - | 34.5 | - |
| InstructBLIP | FLAN-T5 | - | 1212.8 / 291.8 | - | - | - | 63.1 | - | 33.4 | |
| Shikra | Vicuna-13B | - | - / - | 58.8 | - | - | - | - | - | - |
| IDEFICS-80B | LLaMA-65B | - | - / - | 54.5 | - | - | - | - | 36.0 | - |
| LLAVA | Vicuna-7B | - | 807.0 / 247.9 | 34.1 | - | - | 38.5 | - | | |
| LLAVA-1.5 | Vicuna-7B | 18.24 | **1510.75 / 348.21** | **64.3** | 35.3 | 85.87 | 69.61 | 46.07 | 54.38 | 28.08 |
| LLAVA-1.5 + ours | Vicuna-7B | **19.32** | 1474.90 / 325.35 | 61.94 | **37.22** | **86.67** | **69.72** | **46.33** | **59.26** | **30.96** |

Table 2: Performance of various models across different tasks. For tasks that were not evaluated in the original LLAVA-1.5 paper, we used lmms-eval (Bo Li* & Liu, 2024) for evaluation. 'LLAVA-1.5 + ours' refers to the performance of the LLAVA-1.5 model, which has been further fine-tuned using LoRA on our synthesized dataset of 3,000 instances.

## 4.3 ANALYSIS OF EXPERIMENTAL RESULTS

Table 2 shows the performance of the baselines and our method across various benchmarks. The results indicate that, with continued LoRA instruction fine-tuning on only 3k synthesized data, the model significantly outperforms the original LLAVA-1.5-7B on 6 out of 8 tasks, with particularly notable improvements in MMMU, VizWiz (val), and DocVQA, demonstrating the effectiveness of our approach.

## 4.4 IMPACT OF THE NUMBER OF SYNTHETIC DATA PER STRATEGY

| Num | ChartQA | MMMU | TextVQA (val) | VizWiz (val) |
|---|---|---|---|---|
| 0 | 18.24 | 35.3 | 46.07 | 54.36 |
| 1 | 18.40 | 35.87 | 45.73 | 57.26 |
| 2 | 19.00 | 37.11 | 46.16 | 59.19 |
| 3 | 19.32 | 37.22 | 46.33 | 59.26 |

Table 3: Performance of different numbers of synthetic data per strategy. "0" represents the baseline, which refers to the original performance of LLAVA-1.5-7B.

We further conducted experiments by synthesizing different amounts of data for each strategy, ranging from 1 to 3, corresponding to overall dataset sizes of 1k, 2k, and 3k. Table 3 presents the performance of the LLAVA-1.5-7B model on ChartQA, MMMU, TextVQA (validation), and VizWiz (validation). The results demonstrate that for a fixed number of strategies, scaling the dataset by increasing the number of images matched to each strategy can further enhance the model's performance. This suggests that our approach holds the potential to construct large-scale datasets.

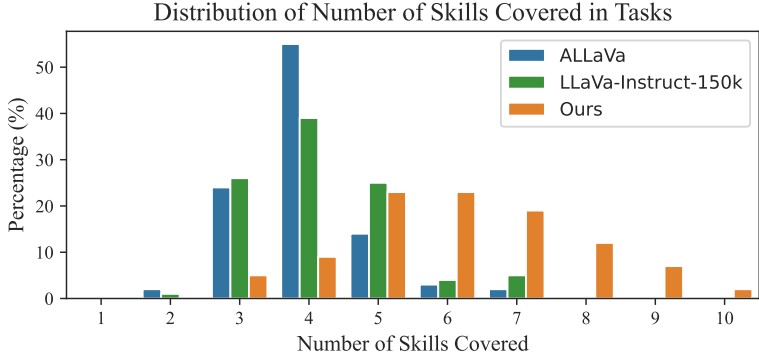

Figure 6: The comparison of three datasets in terms of numbers of different skills covered in each task.

## 4.5 STATISTICAL ANALYSIS OF QUERY COMPLEXITY

We conducted a quantitative analysis of the dataset from the perspective of query complexity. To measure query complexity, we performed a statistical analysis focusing on the number of skills required to solve each sampled problem. The more skills a task involves, the more complex it is. We used GPT-4o for this task, with the prompt template provided in the appendix. Additionally, we randomly selected a subset of synthesized complex reasoning problems from ALLaVa (Chen et al., 2024) and LLaVa-1.5 (Liu et al., 2024a) for comparison, ensuring an equal number of samples.

The Figure 6 shows that instructions from LLaVa-1.5 and ALLaVa cover 2 to 7 skills, with most centered around 4. In contrast, instructions from our dataset exhibit a broader range of skill coverage, spanning from 3 to 10, with a higher overall complexity, most tasks requiring around 6 skills. This suggests that our dataset is intrinsically more complex and diverse than the other two.

| Method | MMMU | SQA(img) | VizWiz (val) |
|---|---|---|---|
| LLAVA-1.5-7B + ours 1k | **35.87** | 69.28 | **57.26** |
| w/o strategy | 35.67 | 69.11 | 56.32 |
| w/o strategy + image matching | 35.44 | 68.85 | 56.08 |
| LLAVA-1.5-7B | 35.3 | **69.61** | 54.36 |

Table 4: Ablation study on MME, SQA(img), and VizWiz (val). "w/o strategy" indicates removing the strategy component from the prompt during data synthesis, primarily ablating the second part of our method. "w/o strategy + image matching" means further ablating the first part of our method by randomly selecting images, essentially evaluating the direct contribution of GPT-4o to model performance.

## 4.6 ABLATION STUDY

We used 1,000 standard synthesized data instances, corresponding to the num=1 setting in Table 3, as the baseline for conducting ablation experiments on the LLava-1.5-7B model, examining the contributions of two key components: (1) The impact of dynamic strategy-based data synthesis on model performance: In this experiment, the strategy is removed from the prompt, with only the image type distribution retained. This isolates the effect of strategies on the overall performance. We used GPT-4 to synthesize 1,000 data instances under this setting. (2) The contribution of GPT-4o to model improvement: We not only removed the strategy mining, but also omitted both retrieval and image filtering. Instead, we randomly downloaded 1,000 images from DataComp-1B and used GPT-4o without strategies to synthesize the instruction data. This experiment fully ablates our method, allowing us to evaluate the direct contribution of GPT-4o to model performance.

Table 4 presents the results across multiple datasets under different settings. The decline in consistency between "LLAVA-1.5-7B + ours 1k" and "w/o strategy" highlights the effectiveness of using dynamic strategies in synthesizing instruction data. The comparison between "w/o strategy" and "w/o strategy + image matching" demonstrates the advantage of retrieving matched images from our curated image library, as opposed to randomly selecting from DataComp-1B. The comparison between 'w/o strategy + image matching' and 'LLAVA-1.5-7B' showcases that the gains provided by GPT-4 alone are relatively limited. In other words, this also demonstrates that when synthesizing data, even with a powerful visual language model, our method can significantly further improve the quality of the synthesized data.

## 5 CONCLUSION

To address the key limitations of existing visual instruction tuning datasets, we introduced a novel strategy-driven approach that synthesizes high-quality instruction data from large-scale image-text pairs. Through a carefully designed pipeline, we automated the process of mining strategies and generating detailed, multimodal instructions tailored to the characteristics of each image. Empirical results demonstrated the effectiveness of our approach, with significant improvements observed after continue fine-tuning using only 3,000 synthesized data samples. The model outperformed LLAVA-1.5-7B across multiple benchmarks, validating the potential of strategy-guided multimodal data synthesis in advancing the performance of vision-language models. For future work, we plan

to investigate the performance of synthesized data on several open-source vision-language models within our framework. Additionally, we aim to secure more computational resources to comprehensively explore the scalability of this approach and its potential for large-scale implementation in industry.

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

## A  APPENDIX

*You are an AI assistant who is good at evaluating the difficulty and complexity of image–text tasks. For each given task and its answer, you should analyse the skill set that is used in this task. Skill sets can be abilities like object detection, mathematic calculation, logical reasoning, etc.. When deciding the involved skill sets for a task, do not use too specific terms like "domain knowledge of fatty acids" or "moment of inertia calculation" but more general and inclusive terms like "chemical knowledge" or "physical calculation". The number of skill sets involved in a task can be diverse and do not feel pressured to give less or more.*
*Here is the task: {}*
*And here is the answer to the task: {}*
*Give your answer directly in the format of following list:*
`["skill set1", "skill set2", "skill set3", ... ]`
*Now give your answer and don't output anything else:*

Figure 7: Prompt used for complexity analysis for a given task.

```bash
#!/bin/bash
deepspeed train_mem.py \
    --lora_enable True --lora_r 128 --lora_alpha 256
    --mm_projector_lr 2e-5 \
    --deepspeed ./scripts/zero3.json \
    --model_name_or_path llava-v1.5-7b \
    --version v1 \
    --data_path
    responses_output_all_domains_filtered_0.65
    _transformed.json \
    --image_folder / \
    --vision_tower openai/clip-vit-large-patch14-336 \
    --mm_projector_type mlp2x_gelu \
    --mm_vision_select_layer -2 \
    --mm_use_im_start_end False \
    --mm_use_im_patch_token False \
    --image_aspect_ratio pad \
    --group_by_modality_length True \
    --bf16 True \
    --output_dir checkpoint/llava-v1.5-7b-task-lora \
    --num_train_epochs 1 \
    --per_device_train_batch_size 16 \
    --per_device_eval_batch_size 4 \
    --gradient_accumulation_steps 1 \
    --evaluation_strategy "no" \
    --save_strategy "steps" \
    --save_steps 50000 \
    --save_total_limit 1 \
    --learning_rate 2e-4 \
    --weight_decay 0. \
    --warmup_ratio 0.03 \
    --lr_scheduler_type "cosine" \
    --logging_steps 1 \
    --tf32 True \
    --model_max_length 2048 \
    --gradient_checkpointing True \
    --dataloader_num_workers 4 \
    --lazy_preprocess True \
    --report_to wandb
```

Figure 8: Command and parameters used in our LoRA finetuning process.

**Art**: artist, bar, interval, art, style, key, pitch, design, building, melody, title, signature, structure, music, figures, century, church, chord, subject, artists, intervals, chords, clef, author, feature, sculpture, patron, clarinet, line, ...

**Business**: price, company, stock, value, costs, sales, cash, income, production, units, bond, portfolio, tax, debt, shares, product, balance, share, project, distribution, labor, growth, economy, inventory, curve, dollars, bonds, investment, business, ...

**Medicine**: body, disease, diagnosis, screening, patients, cases, cancer, examination, subjects, blood, appearance, population, thalassaemia, exposure, risk, incidence, age, cell, vaccine, heart, hospital, reaction, time, serum, health, test, structure, pressure, ...

**Science**: reaction, pressure, area, structure, force, gas, length, value, foundation, water, points, field, region, energy, order, sample, mass, compound, angle, solution, temperature, function, axis, distance, wire, level, cell, data, section, circuit, change, resistance, weight, direction, circle, statements, speed, ...

**Technology**: tree, node, code, stress, diameter, water, force, pressure, circuit, flow, steel, system, pipe, velocity, heat, point, tank, temperature, power, plate, mass, bar, rod, unit, shaft, gas, terms, plane, steam, weight, state, speed, voltage, pin, strain, link, tube, volume, spring, network, turbine, ...

**Sociology**: study, group, participants, brain, treatment, memory, stress, symptoms, language, system, research, behavior, researcher, response, factors, studies, disorder, sleep, drug, therapy, levels, control, nerves, movement, axis, health, experiment, patient, behaviors, individual, percentage, mortality, ...

Figure 9: Visualization-related keyword examples for different domains.

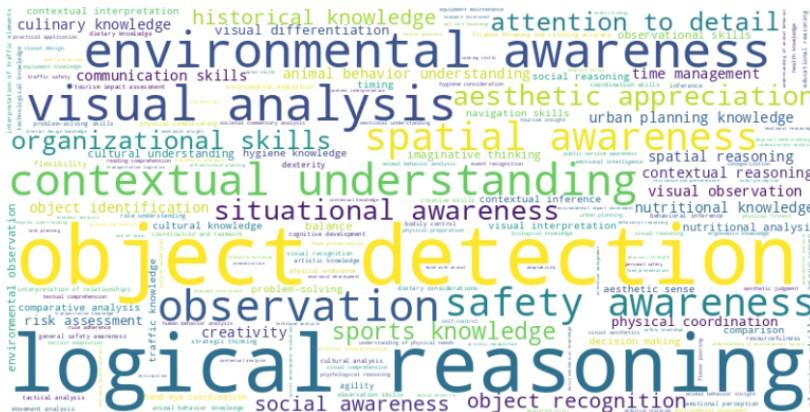

Figure 10: The wordcloud of the skill sets covered in tasks from LLaVa-1.5 complex reasoning dataset.

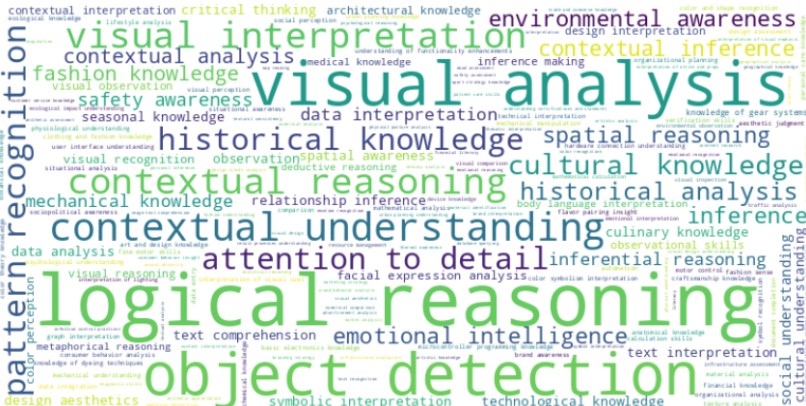

Figure 11: The wordcloud of the skill sets covered in tasks from ALLaVa complex reasoning dataset.

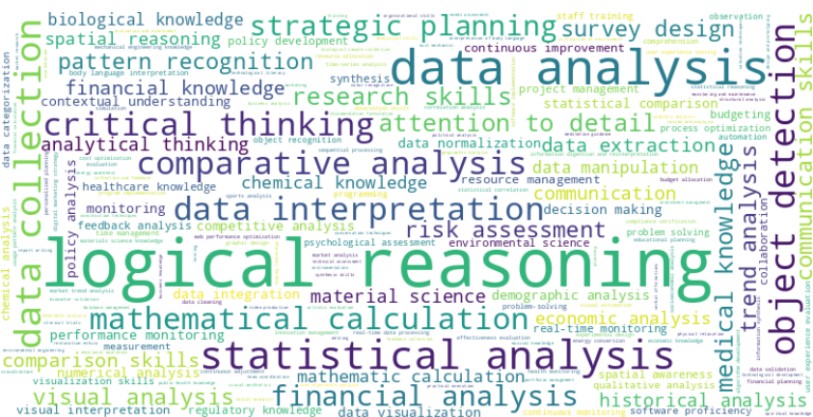

Figure 12: The wordcloud of the skill sets covered in tasks from our dataset.

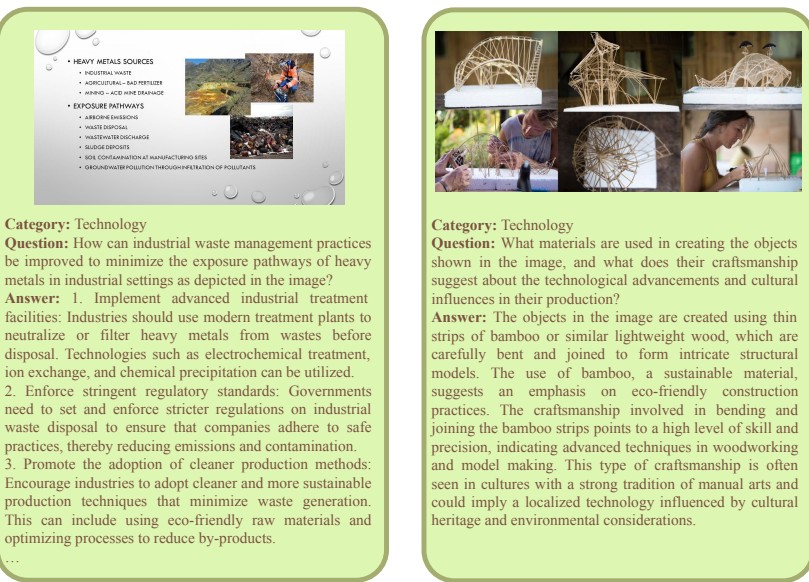

Figure 13: Examples of synthesized instruction of Technology domain from our dataset.

918
919
920
921
922
923
924
925
926
927
928
929
930
931
932
933
934
935
936
937
938
939
940
941
942
943
944
945
946
947
948
949
950
951
952
953
954
955
956
957
958
959
960
961
962
963
964
965
966
967
968
969
970
971

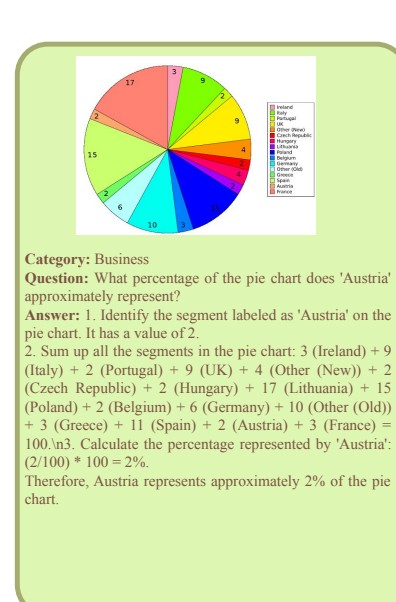
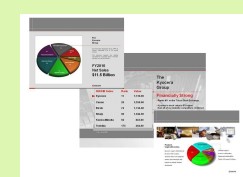

**Category:** Business
**Question:** What percentage of the pie chart does 'Austria' approximately represent?
**Answer:** 1. Identify the segment labeled as 'Austria' on the pie chart. It has a value of 2.
2. Sum up all the segments in the pie chart: 3 (Ireland) + 9 (Italy) + 2 (Portugal) + 9 (UK) + 4 (Other (New)) + 2 (Czech Republic) + 2 (Hungary) + 17 (Lithuania) + 15 (Poland) + 2 (Belgium) + 6 (Germany) + 10 (Other (Old)) + 3 (Greece) + 11 (Spain) + 2 (Austria) + 3 (France) = 100.\n3. Calculate the percentage represented by 'Austria': (2/100) * 100 = 2%.
Therefore, Austria represents approximately 2% of the pie chart.

**Category:** Business
**Question:** How should Kyocera strategically plan its resource allocation to optimize its net sales distribution across different sectors, given the FY2010 net sales data and competitor rankings?
**Answer:** 1. Analyzing Current Allocation:
 - Analyze the existing net sales distribution across sectors: Electronics Devices (12%), Applied Ceramics (11%), Telecommunications (7%), Information Equipment (55%), Other (4%), Fine Ceramics (7%), Semiconductor Parts (4%).
2. Identify Growth Opportunities:
 - Conduct market analysis to identify high-growth potential sectors.
 - Evaluate competitor strengths and weaknesses based on NIKKEI Index rankings.
3. Prioritize High-Potential Sectors:
 - Allocate more resources to high-potential sectors such as Information Equipment. …

Figure 14: Examples of synthesized instruction of Business domain from our dataset.

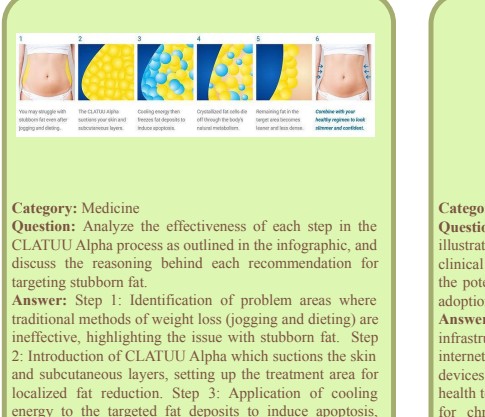
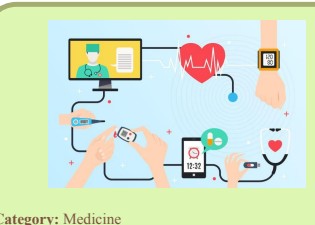

**Category:** Medicine
**Question:** Analyze the effectiveness of each step in the CLATUU Alpha process as outlined in the infographic, and discuss the reasoning behind each recommendation for targeting stubborn fat.
**Answer:** Step 1: Identification of problem areas where traditional methods of weight loss (jogging and dieting) are ineffective, highlighting the issue with stubborn fat. Step 2: Introduction of CLATUU Alpha which suctions the skin and subcutaneous layers, setting up the treatment area for localized fat reduction. Step 3: Application of cooling energy to the targeted fat deposits to induce apoptosis, which is a natural process of programmed cell death, aimed at reducing fat cells. Step 4: Representation of the effect of the cooling energy on fat cells, where the fat cells become crystallized and start dying, to be metabolized by the body. Step 5: Demonstration of the reduced density of fat in the targeted area post-treatment, indicating the effectiveness of the process in making the area leaner. …

**Category:** Medicine
**Question:** How can connected health technologies, as illustrated in the image, be effectively implemented in a clinical setting to improve patient outcomes, and what are the potential benefits and challenges associated with their adoption?
**Answer:** 1. Assess the current state of technology and infrastructure in the clinical setting. Ensure adequate internet connectivity, secure data storage, and compatible devices. 2. Identify specific patient needs that connected health technologies can address, such as remote monitoring for chronic diseases or telemedicine consultations. 3. Integrate connected health devices into routine clinical workflows, such as wearables for monitoring vital signs, connected medication dispensers, and telehealth platforms for remote consultations. 4. Train healthcare providers on the use, maintenance, and troubleshooting of these technologies, emphasizing data privacy and patient consent. …

Figure 15: Examples of synthesized instruction of Medicine domain from our dataset.

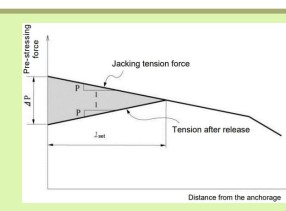

**Category:** Science
**Question:** Given the diagram showing the relationship between pre-stressing force and distance from the anchorage, how would you decompose the jacking tension force (P) into its horizontal and vertical components? Additionally, calculate the change in pre-stressing force (ΔP) over a distance (L_set).
**Answer:** To decompose the jacking tension force (P) into its horizontal (P_x) and vertical (P_y) components, we can use the following steps:
1. Identify the angle of inclination (θ) of the jacking force. This can be found through trigonometric relationships given in the diagram.
2. Use trigonometric functions to decompose the force:
P_x = P * cos(θ), P_y = P * sin(θ)
Next, calculate the change in pre-stressing force (ΔP) over the distance L_set:
1. From the diagram, ΔP is the difference between the initial jacking tension force and the tension …

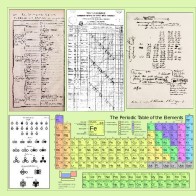

**Category:** Science
**Question:** Observing the image, classify the elements in the periodic table based on their atomic numbers into three categories: low (1-20), mid (21-40), and high (41 and above). What chemical properties and trends can you identify for each category?
**Answer:** To classify the elements, follow these steps:
… For each category, observe the following trends and properties:
- Low category (1-20): These elements include nonmetals, metals, and metalloids. They display a range of properties from highly reactive (alkali metals and halogens) to noble gases (inert). They tend to form simple binary compounds.
- Mid category (21-40): These elements are primarily transition metals. They show properties like high melting points, formation of colored compounds, and variable oxidation states. They are often good catalysts and form alloys. …

Figure 16: Examples of synthesized instruction of Science domain from our dataset.

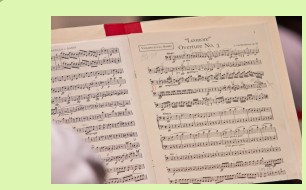

**Category:** Art
**Question:** Analyze the use of dynamics and articulation in the first two lines of the 'Leonore' Overture No. 3 for Violoncello e Basso. How do these elements contribute to the musical expression and overall structure?
**Answer:** In the first two lines of the 'Leonore' Overture No. 3 for Violoncello e Basso, we observe a range of dynamics including pianissimo (pp), fortissimo (ff), and crescendos. The use of articulation markings such as staccato and accents provide contrast and emphasis on certain notes. The opening measures start with a soft, delicate texture indicated by pianissimo, creating a sense of tension and anticipation. The crescendos lead to louder dynamics, increasing the intensity and driving the music forward. The staccato notes add a sense of urgency and movement, while the accented notes highlight important melodic or harmonic points. Together, these elements shape the musical expression, guiding the listener through contrasting moods and contributing to …

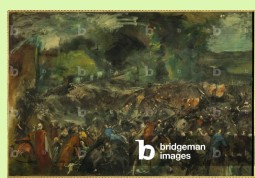

**Category:** Art
**Question:** How does the artist use color, composition, and subject matter to convey the intensity and chaos of the historical battle depicted in the painting?
**Answer:** Step 1: Analyze the use of color in the painting. The artist uses dark and muted tones to establish a somber and chaotic atmosphere, which is characteristic of battle scenes. The varying shades of red, brown, and dark green emphasize the violence and confusion of the battlefield. Step 2: Examine the composition. The painting is densely packed with figures and elements, with no clear focal point, contributing to a sense of disorder and movement. The composition directs the viewer's eye to different parts of the scene, preventing the viewer from settling on one spot, mirroring the erratic nature of battle. Step 3: Consider the subject matter. The painting likely depicts a historical battle, suggested by the presence of soldiers in uniforms, horses, and weapons. The figures are engaged in combat, their postures and actions illustrating the struggle …

Figure 17: Examples of synthesized instruction of Art domain from our dataset.

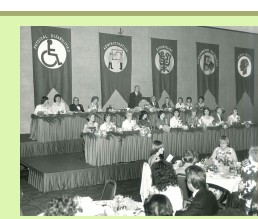

**Category:** Sociology
**Question:** What can be inferred about the roles and hierarchies of individuals in the group based on their seating arrangement and positioning at the event depicted in the image?
**Answer:** To infer the roles and hierarchies of individuals based on their seating arrangement, begin by identifying those seated at the head table, which suggests a position of prominence. The individual standing behind the podium likely holds a leadership or speaker role, indicating a higher status. The central positioning of certain individuals at the head table compared to those at the ends could signify higher importance or rank. Additionally, notice how body language and attention are directed towards the podium, indicating respect or subordination. Evaluating attire and engagement levels can also provide hints about the differing roles, with more formally dressed individuals potentially holding higher positions.

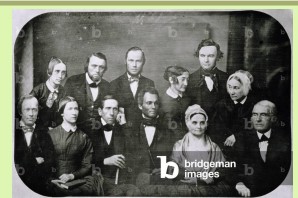

**Category:** Sociology
**Question:** Identify the historical context of this 19th-century photograph. Discuss the significance of the attire and the likely social class of the individuals, referring to similar examples of famous figures or events from that era.
**Answer:** This photograph appears to be from the 19th century, judging by the attire and the style of the photograph. The individuals are dressed formally, indicating that they are likely from the middle or upper class. During the 19th century, clothing was a significant indicator of one's social status. Men are wearing three-piece suits, which were common among business professionals and the wealthy. Women are dressed in modest, high-collared dresses, which was typical for women of higher socioeconomic status who followed strict social norms. A pertinent example of similar attire can be seen in photographs of famous figures like Charles Dickens or members of the American Civil War era upper class. The individuals in this image could be part ...

Figure 18: Examples of synthesized instruction of Sociology domain from our dataset.

