# OpenReview forum: "Strategy-centric Synthesis: Connecting Billions of Image-Text Pairs to High-Quality Visual Instruction Data"
_ICLR.cc/2025/Conference — ICLR 2025 Conference Withdrawn Submission_

### Official Review · Reviewer_zAAu · 2024-10-27

**Soundness:** 2
**Presentation:** 3
**Contribution:** 2
**Rating:** 5
**Confidence:** 3

**Summary:**

The paper proposes a strategy-centric synthesis approach for generating high-quality visual instruction data from large-scale image-text datasets, addressing limitations in diversity, complexity, and alignment with real-world queries in current visual instruction datasets. By mining complex, domain-specific images and applying tailored query strategies, this method generates high-quality questions and answers that better reflect real-world multimodal tasks. The synthesized data significantly improves the performance of the LLAVA-1.5 model on multiple benchmarks, demonstrating the effectiveness of the approach.

**Strengths:**

1. The paper presents a unique strategy-centric paradigm, moving beyond traditional task-based instruction tuning by emphasizing diverse and complex query strategies tailored to specific image characteristics.

2. The inclusion of self-reflection for evaluating response quality is an interesting idea for improving the reliability of synthesized instructions.

**Weaknesses:**

1. The evaluation would benefit from more extensive comparisons with existing synthesis techniques, particularly those that use multimodal datasets, e.g., LVIS-Instruct4v and ShareGPT4v.

2. The comparison in Table 2 seems a bit unfair, it would be better to re-train the LLaVA 1.5 model by replacing LLaVA-558-Instruct with the proposed SFT data.

**Questions:**

Please refer to the weakness part.

---

### Official Review · Reviewer_uQag · 2024-10-31

**Soundness:** 3
**Presentation:** 3
**Contribution:** 2
**Rating:** 5
**Confidence:** 5

**Summary:**

The paper introduces "Strategy-Centered Synthesis," a novel method for generating high-quality visual instruction data to overcome limitations in current datasets. Key Contributions:

Automated Strategy Mining: Extracts detailed queries from seed images, enabling better question generation for complex scenes.

High-Quality Seed Image Library: Builds a diverse library to enhance data quality.

Experimental Validation: Demonstrates significant performance improvements in benchmarks using a small fraction of the LLAVA-1.5 dataset.

**Strengths:**

The paper's strengths lie in its introduction of the "Strategy-Centered Synthesis" method, which automatically generates high-quality visual instruction data, effectively addressing the limitations of existing datasets. By employing automated strategy mining and a diverse seed image library, the method enhances the granularity and complexity of question generation. Experimental results demonstrate significant performance improvements in model benchmarks using a small number of newly synthesized samples.

**Weaknesses:**

The main drawback of the paper lies in its experimental setup, where the authors validate the method's effectiveness by further fine-tuning LLAVA-1.5 using LoRA on a synthesized dataset of only 3,000 instances, which is considered insufficient for robust evaluation.

The reasons for considering the experimental setup insufficient are twofold:

An overly simplistic experimental design may lead to unavoidable fluctuations, meaning the performance gains observed from the LoRA + 3,000 samples setup could be attributed to training variability rather than the quality of the synthetic data. The authors can reduce training variance through the following method: conducting multiple training runs and averaging the results. For example, both the baseline and the models trained with additional data could be trained five times, and the average performance along with variance could be reported in the results. This approach provides a more accurate reflection of the model’s actual performance and minimizes the impact of variance from individual training runs, allowing for a more scientific assessment of the effectiveness of data augmentation.

The aim of developing data synthesis pipelines is to enable fully automated large-scale production of high-quality data; thus, performance evaluation should only occur when the pipeline generates a sufficient quantity of data. Too few samples might result in performance improvements that are simply due to variability during the data synthesis process. I believe the authors need to generate at least 50k samples to properly assess the effectiveness of the strategy-centered data synthesis method at scale. In addition to incremental training, I suggest replacing part of the original LLAVA SFT data generated by GPT-4V with the newly synthesized instruction data for a direct quality comparison. This replacement experiment would allow for a more accurate evaluation of the actual improvement brought by the synthetic data, as it would reduce the interference of incremental training effects.

**Questions:**

I recommend that the authors reconsider the experimental setup to ensure a fair and comprehensive evaluation. Specifically, it would be beneficial to increase the dataset size to at least 50,000 instances to mitigate performance fluctuations and allow for meaningful comparisons. Additionally, aligning this setup with other synthesis-based methods, such as LLAVA-1.5 SFT data, Allava, and MMInstruct, would enable a fair assessment of the proposed method’s effectiveness.

---

### Official Review · Reviewer_CatY · 2024-11-01

**Soundness:** 3
**Presentation:** 3
**Contribution:** 2
**Rating:** 3
**Confidence:** 4

**Summary:**

This paper introduces a new approach to enhance Vision-Language Models (VLMs) by addressing gaps in current visual instruction tuning datasets. Traditional datasets often limit question variety, image diversity, and task complexity, hindering VLMs' ability to tackle real-world queries. To overcome these limitations, the authors propose a strategy-centric synthesis method that generates diverse, high-quality instruction data. Using DataComp-1B image-text pairs, complex images are filtered and paired with targeted prompts, creating enriched multimodal instructions. Empirical results reveal that fine-tuning with only 3,000 synthesized samples significantly boosts model performance across benchmarks, demonstrating the method's potential for broadening VLM capabilities.

**Strengths:**

1. The strategy-centric synthesis method enhances data diversity and realism, improving VLM versatility.
2. Fine-tuning with only 3,000 samples improves model performance across benchmarks.
3. The organization of this paper is logical.

**Weaknesses:**

1. The motivation presented in the paper lacks persuasiveness, particularly regarding the claim that existing multimodal instruction data lacks **diversity** and **complexity**. First, regarding diversity, current multimodal instruction benchmarks like MMMU [1] and MMBench [2] already draw from various domains and tasks, with well-structured diversity classifications. As for complexity, existing multimodal data is not limited to basic visual tasks like captioning or OCR; some instruction data requires models to solve complex scientific problems and perform fine-grained perception, demanding comprehensive capabilities. It would strengthen the authors’ argument if they provided specific statistical data to support these claims and better justify their motivation.
2. The writing in the methods section needs improvement, particularly in how the proposed prompts effectively address the previously identified gaps in question variety, image diversity, and task complexity. The paper presents a general overview of the approach but lacks specific details on how the prompts are designed to fully leverage these aspects.
3. One of my main concerns: what is the detailed strategy for dealing with generated noisy or bad instruction samples?


_**Ref**_:

[1] MMMU: A Massive Multi-discipline Multimodal Understanding and Reasoning Benchmark for Expert AGI. In CVPR 2024.

[2] MMBench: Is Your Multi-modal Model an All-around Player? In ECCV 2024.

**Questions:**

Please refer to the Weaknesses.

---

### Official Review · Reviewer_TdJL · 2024-11-04

**Soundness:** 1
**Presentation:** 3
**Contribution:** 2
**Rating:** 3
**Confidence:** 5

**Summary:**

This paper proposes a new method for converting traditional image-text pairs into instruct data, and the effectiveness of the method has been validated through experiments.

**Strengths:**

1. This paper assists the VLM model in generating instruct data based on images by constructing suitable strategies, and the motivation behind the method is reasonable.
2. After using the newly generated data, the performance of LLAVA has been improved.

**Weaknesses:**

1. My main concern is the effectiveness of the method proposed in this paper on large-scale data experiments. This is because the strategy is not only related to the domain but also closely related to the content of the images. A universal strategy may not necessarily be effective when producing instruct data on a large scale. Moreover, using the same strategy can lead to the generation of similar data, resulting in reduced diversity.
2. Additionally, the paper lacks comparisons with other methods, including different data generation techniques and existing datasets. It is possible that simply introducing some heterogeneous data could yield similar performance improvements as those reported in this paper, thus the effectiveness of the method proposed in this paper is not solid.

**Questions:**

1. What is the trend of the model's performance improvement as the amount of data increases?
2. The experimental results for starting SFT from scratch, rather than continuing training, are missing.
3. Comparisons with other methods and datasets.

---

### Official Review · Reviewer_F7J3 · 2024-11-04

**Soundness:** 3
**Presentation:** 2
**Contribution:** 2
**Rating:** 3
**Confidence:** 5

**Summary:**

This paper introduces a novel paradigm called "strategy-centric synthesis," which automatically synthesizes high-quality instruction data from large-scale image-text pairs to address the limitations of existing visual instruction tuning datasets. By automating strategy mining and strategy-guided multimodal instruction synthesis, the method generates more diverse, complex, and high-quality visual instruction data. Experimental results demonstrate that fine-tuning with only 3,000 newly synthesized data samples outperforms the original LLAVA-1.5-7B across multiple benchmarks, validating the effectiveness of this approach.

**Strengths:**

1. Novel Paradigm: The paper introduces a strategy-centric synthesis paradigm that leverages large-scale image-text pairs to generate high-quality visual instruction data, addressing the limitations of current datasets derived from past visual tasks.
2. Effectiveness and Scalability: The proposed method demonstrates significant improvements in model performance across multiple benchmarks with only a small fraction of synthesized data, showcasing its effectiveness and potential for scalability in high-quality data synthesis.

**Weaknesses:**

In my opinion, the key to building an SFT data production pipeline lies in demonstrating the **completeness** and **scalability** of the method, rather than merely proving its **effectiveness**. This is because, in the SFT stage, many methods can be found to have improved metrics by expanding diversity at various levels.
Completeness represents: new tasks in a large number of open domains can be covered or involved as the method proposed in this paper scales up. However, this paper only validates this on specific fixed evaluation benchmarks e.g., MMBench/MMMU/POPE/VizWiz, which is not a sufficient argument. The article also lacks the verification of building data diversity.
Scalability represents: this method can achieve scaling up in terms of task or prompt diversity by scaling up the data source or a certain algorithmic step. This paper also lacks discussion on this characteristic.

Therefore, I believe this paper is incomplete.

**Questions:**

I suggest that the authors add more discussions about the completeness and scalability of the method presented in this paper, as well as include more human studies on results, rather than relying solely on benchmarks. This is because benchmarks themselves are not entirely reliable.

---

### Note · Authors · 2024-11-15

**Comment:**

Thank you for the reviewers' comments. In fact, before the rebuttal period, we had already made some improvements addressing several concerns that were also raised by the reviewers, such as experimental setup, data scale, comparison with baseline methods, and more. However, after careful consideration, we have ultimately decided to withdraw the paper. Thank you again for your valuable feedback.

**Withdrawal Confirmation:**

I have read and agree with the venue's withdrawal policy on behalf of myself and my co-authors.